# The Relationship of Rejection Sensitivity to Depressive Symptoms in Adolescence: The Indirect Effect of Perceived Social Acceptance by Peers

**DOI:** 10.3390/bs14010010

**Published:** 2023-12-22

**Authors:** Theodoros Giovazolias

**Affiliations:** Laboratory of Applied Psychology, Department of Psychology, University of Crete, 74100 Rethymnon, Greece; giovazot@uoc.gr

**Keywords:** rejection sensitivity, depressive symptoms, perceived social acceptance, peer relationships, adolescence, indirect effect

## Abstract

Rejection sensitivity (RS), the tendency to expect, perceive and overreact to rejection from others, has been linked to children’s and adolescents’ increased vulnerability to depressive symptoms, negatively affecting their perceptions of the quality of their relationship with their peers. The aim of this study is to examine (a) the indirect effect of perceived peer social acceptance in the relationship between RS and depressive symptoms in a sample of Greek adolescents, (b) the differential effect of the different components of rejection sensitivity (angry and anxious RS) on the model, and (c) possible gender differences. The sample of the study consists of 295 adolescents (139 boys, 156 girls, M_age_ = 14.20, *SD* = 1.60) residing in the greater Heraklion Prefecture area. Data collection was carried out using self-report questionnaires that measured demographic characteristics, self-perceptions about their peer relationships (self-perception profile for adolescents/SPPA), susceptibility to rejection (children’s rejection sensitivity questionnaire/CRSQ), and depressive symptoms (children’s depression inventory/CDI). Results showed that (a) RS was positively related to depressive symptoms and negatively related to adolescents’ perceptions of their relationships, (b) relationship perceptions were negatively related to depressive symptoms, and gender differences were also found, (c) perceived social acceptance by peers was found to have an indirect effect in the relationship between RS and depressive symptoms, with girls showing a greater effect, and (d) differences were observed in the mediating model between the components of RS, with the mediating effect of perceptions being higher in the model with anxious RS, which appears to confirm previous theoretical postulations. The results of this study highlight the importance of adolescents’ perceptions of their peer relationships in the occurrence of depressive symptoms during this developmental period, especially in youths with anxious rejection sensitivity.

## 1. Introduction

Psychological research in social and psychological development has drawn attention to the importance that close interpersonal relationships play in an individual’s psychological safety and well-being [1,2,3]. During childhood, relationships with parents play the most important role, while social experiences with peers become more important progressively from early adolescence, where individuals spend more and more time outside the family environment, focus on their peer group, and generally make greater efforts to belong [4]. Although the desire to be socially accepted and to belong to groups is so universal, as it is considered a fundamental human motivation that motivates interpersonal relationships at all ages [5], adolescence is the period when experiences of rejection are a particularly stressful event. Toward this end, research has extensively shown that adolescents’ psychological and social adjustment depends on the extent to which relationships with peers and friends are perceived as exclusive and accepting, rather than rejecting [6,7]. This focus and anxiety towards social acceptance can make adolescents vulnerable by keeping them alert for signs of rejection, to which they might react in a variety of ways, showing, for example, anger, anxiety, dependency, and jealousy, as well as internalizing problems such as depressive symptoms [8,9].

This study examines the overarching hypothesis that perceived social acceptance by peers has an indirect effect in the relationship between rejection sensitivity (RS)and depressive symptoms in an adolescent sample.

Diathesis–stress models suggest that the development of psychopathology results when an individual with a certain vulnerability is exposed to a given set of environmental conditions. More specifically, it is proposed that when a diathesis is combined with a life stressor (e.g., relational rejection), individuals become more vulnerable for developing psychological difficulties [10]. Current diathesis–stress models of depression point out that both negative cognitive styles and relational factors may predispose individuals to experience more depressive symptoms [11].

### 1.1. Rejection Sensitivity (RS)

In the relevant literature, the tendency to overreact to even small or ambiguous signs of rejection has been defined as rejection sensitivity (RS), a conceptual construct that refers to the cognitive and emotional predisposition to defensively expect and perceive rejection from others, either with anxious (anxious RS) or angry expectations of rejection (angry RS) [8]. RS is characterized by negative interpretations of social circumstances, which potentially lead to negative responses to others’ behaviour, negative social interactions, resulting in negative emotions [12]. Downey et al. [13] have stressed that anxious RS would be more strongly linked with internalizing difficulties such as depressive symptomatology and social anxiety, whereas angry RS would be more strongly associated with externalizing difficulties, such as aggressive behaviours. Further, anxious RS has been linked with other psychological difficulties such as social anxiety and ADHD, especially in adolescent samples [14].

RS has been proposed as a stable, individual trait, which in itself may help explain the development of internalizing disorders such as depression. Several studies provide evidence that RS might best be characterized as an individual vulnerability factor that increases one’s risk of developing depression in the context of specific stressors [15]. A cross-sectional study by McDonald et al. [16] found an association of RS with depressive symptoms in adolescents, but only in those adolescents who showed low support from family and friends. Although most research on the link between RS and depression has focused on anxious expectations of rejection, as it is anxious RS that is closely associated with feelings of sadness as the central outcome of depression [17], the association of angry RS with depressive symptoms should not be ignored. Expectations of rejection characterized by anger have been found to be closely associated with externalizing problems [9,12], while little or almost no research has been conducted on their association with depressive symptoms [18]. However, this seems important given that depressive symptoms and aggressive behaviour appear to be closely related [19,20], particularly in adolescent samples [21], while anger is associated with both childhood and adolescent depression [22]. Recent research has confirmed that the constant anticipation of rejection—whether anxious or angry—creates feelings of anxiety that promote depressive symptoms [23,24]. Therefore, it is assumed in the present research that angry RS could equally predict depressive symptoms.

### 1.2. Indirect Effect of Peer Acceptance

While previous research has demonstrated a link between anxious and angry anticipation of rejection at both cross-sectional and longitudinal designs [25,26], little is still known about the mechanisms by which RS is a potential risk for depressive symptoms in adolescents. One possible mechanism linking susceptibility to rejection and depression is peer relationships and, more specifically, perceived acceptance by peers. Acceptance reflects adolescents’ relative liking of their peer group [27]. As noted earlier, the need to belong and the need to bond with peers becomes particularly important during adolescence. This need emerges through the change in relationships due to the transition to secondary education, independence from parents, and the onset of romantic relationships. The interpersonal risk model [28] posits that negative peer relationships are a significant stressor that contributes to maladaptive outcomes such as depressive symptoms. A wealth of research appears to support this hypothesis. For example, Papadaki and Giovazolias [29] and Israel and Gibb [30] report that peer victimization predicts depressive symptoms in adolescents cross-sectionally, both a year later and in the long term.

Although being liked by peers has been linked to adolescents’ mental health, the significant contribution of perceived social competence and acceptance by others, or more generally, adolescents’ cognitive representations of their social relationships (otherwise referred to as interpersonal schemas) [31], have been identified as more direct causes of depressive symptoms in cognitive sensitivity models [32]. The importance of self-perceived competence has been supported by numerous studies [33,34] and appears to be negatively related to poor adjustment and psychological symptoms, while also being associated with the quality and quantity of social interactions [35].

In longitudinal studies of children in the transition to adolescence, relatively low perceptions of social competence have also been associated with later-developed depression [36]. Previous research [37] in adolescents has linked negative self-assessments of social competence with high levels of depressive symptoms, social anxiety, and social avoidance behaviours. Other researchers [38] have highlighted the mediating role of perceived social acceptance by peers, which means that adolescents’ own perceptions of their own lack of acceptance, rather than their actual lack of liking/acceptance by their peers, appears to be more directly related to depressive symptomatology.

### 1.3. Gender Differences

Mixed gender differences in levels of RS emerge early in adolescence, with boys being less interpersonally oriented [38], placing more emphasis on group activities with peers over closeness and disclosure [39], and having fewer close friendships than girls [40]. As a possible result, boys may experience more discomfort in forming cross gender relationships [41]. As adolescent boys experience more social pressure in initiating romantic relationships, they also face an increased likelihood of being publicly rejected [42]. It therefore appears that boys are particularly affected by social forms of rejection [43].

However, another line of research has emphasized the socially constructed relational direction of female identity, assuming that high rejection sensitive young females will be more likely to experience interpersonal distress such as a higher fear of intimacy compared to males [44]. Indeed, it has been supported that females appear to be more sensitive to social stimuli than males and are more likely to harm their relationships as a response to interpersonal stressful incidents [45] and that late female adolescents [46] as well as young female adults [39] report significantly greater RS and interpersonal anxiety than males. Other research has shown that male adolescents are more sensitive to signs of rejection when their social status is threatened [9], and this sensitivity combined with relatively underdeveloped emotional-supportive relationships may explain the higher levels of RS in males [47]. However, in a relatively recent meta-analysis, no gender differences in levels of RS were found [48].

Gender differences in depression become apparent at the onset of adolescence, between 13 and 15 years of age, and continue to increase in late adolescence (16–19 years). According to a previous study by Angold and Rutter [49], in a large clinical sample, the rate of depressive disorders and symptoms was found to be similar in boys and girls before the age of 11 years, but from 14 to 16 years, girls were twice as likely as boys to have depressive symptoms. Similarly, other research has showed small gender differences from age 13 to 15 years, with the largest differences occurring at 15 and 18 years [50]. Despite the variation in the occurrence of diagnosed depression and depressive symptoms across the lifespan, sex differences continue to have remarkable consistency through until at least age 55 [51].

The research findings regarding gender differences in perceived peer acceptance are also mixed. For example, males and females vary in how they perceive peer acceptance, as boys show a greater inaccuracy for perceived acceptance by same-sex peers compared to girls, while girls seem to have a more negatively biased perception of opposite-sex acceptance compared to boys [52]. In a recent systematic review [53], it was found that gender moderated the effects of actual social status on scholastic achievement, with adolescent females being more affected by poor acceptance. Further, Bédard et al. [37] found that girls may show vulnerability to peer acceptance at an earlier age than boys. Overall, some studies have reported that girls typically receive higher scores for peer acceptance and perceived popularity than boys [27], whereas others have reported no gender differences in perceived peer acceptance [54]. Because of these mixed findings, the role of gender was explored in all analyses without any specific hypotheses. The present study aims to investigate the relationship between RS and the occurrence of depressive symptoms in a Greek adolescent population. It should be noted here that depression symptomatology represents a major health issue for Greek adolescents as its prevalence has been found to be higher compared to similar age groups across many European countries [55]. Furthermore, Greece has entered a long period of economic crisis, especially after the break of COVID-19 pandemic with major adverse effects on many areas of the life of the population. We therefore carefully consider the investigation of some significant mental health aspects of Greek adolescents during this difficult period.

As previously reported, RS appears to affect individuals’ cognitive perceptions, and in particular how they perceive their relationships as acceptable or rejecting. Further, relevant research indicates that individuals with high RS are more likely to exhibit high arousal in their interpersonal relationships, often leading to heightened experience of interpersonal distress [56]. For example, it has been suggested that people who are highly sensitive to rejection tend to avoid social interactions in an effort to protect themselves from experiencing feelings of rejection and/or other unpleasant feelings (e.g., loneliness, fear of intimacy, etc.) [57]. It can be assumed then that these individuals may exhibit anxious representations and interaction patterns in their social relationships [58] which may distort/impair their perceived social acceptance by peers. For these reasons, we included RS as an independent variable in our model that would predict perceived social acceptance. Also, perceived social acceptance in peer relationships, rather than peer relationships per se, plays an important role in adolescents’ psychological health, and appears to be related to depressive symptomatology. For this reason, in addition to the impact that susceptibility to rejection appears to have on the onset of depressive symptoms, the indirect effect of adolescents’ perceived peer acceptance on this relationship is also examined. In this paper, the research hypothesis is based on the interpersonal risk model, where the adolescent’s relationships with peers and, more specifically, the adolescent’s perceptions of interpersonal relationships are what predict the onset of depressive symptoms.

## 2. Materials and Methods

### 2.1. Sample

The total sample of the study consisted of 295 high school students aged 13–15 years (Mean_age_ = 14.2 years and *SD* = 1.6), randomly selected from different areas of Heraklion Prefecture, Crete. The percentage of females is 53% (156), while the percentage of males is 47% (139).

### 2.2. Procedure

Participation in the study required parental consent and adolescent assent. A sealed letter was sent to each parent through their children with a detailed description of the nature and goals of the study inviting them to provide their written consent. Children who did not obtain written permission from parents were excluded from participating in the study. There were no other exclusion criteria for taking part in this study. The parental consent rate was 93%, with most of the nonparticipants simply failing to return consent forms.

The administration of the questionnaires took place in the participants’ classrooms during ordinary class sessions and lasted approximately 30 min. Issues of anonymity and confidentiality were particularly highlighted, along with the voluntary participation in the study. Permission to conduct the research was granted by the Departmental Ethics Committee. There was no payment or other incentive to complete the questionnaires. After the completion of the questionnaires, adolescents were debriefed and thanked for their participation in the study. The data were collected during the 2022 spring semester. Eventually, they were encoded, transferred, and analysed with SPSS 25 at the University of Crete.

### 2.3. Plan of Analysis

The SPSS (Statistical Package for Social Sciences, version 25.0, University of Crete) was used for the data analysis. We tested gender differences in the variables using *t*-test for independent samples, and the correlations among study variables through Pearson correlation. We conducted a priori power analysis using the software package, G*Power 3.1 [59]. The analysis indicated that a sample size of 140 would be sufficient to detect significant meditational effects with a power of 0.95 and an alpha of 0.05. However, in order to increase the power of the tests used to examine our hypotheses and hence the overall power of this study, we opted for collecting a greater sample size (i.e., 295 students). The hypothesis that perceived peer acceptance would have an indirect effect on the relationship between rejection sensitivity and depressive symptoms was examined using PROCESS 3.1 version [60]. We followed Baron and Kenny [61] recommendations regarding the design of a mediation model, according to which the independent variable is required to predict the mediator and the mediating variable is required to predict the dependent variable. We used a bias corrected bootstrap technique with 5000 samples and a set of 95% confidence interval.

### 2.4. Measures

Demographics. Participants filled out a demographic questionnaire with information about their gender, age, family status, education of their parents, and grade of study.

*Rejection Sensitivity*. The Children’s Rejection Sensitivity Questionnaire (CRSQ) was developed by Downey et al. [13] and adapted by Spiropoulou and Giovazolias [62] for Greek youth and adolescents. The scale covers twelve scenarios, such as: “*Imagine you’re in the bathroom at school and you hear your teacher in the hallway outside talking about a student with another teacher. You hear her say that she really doesn’t like having this child in her class. You wonder if she could be talking about you*.” For a total of 36 items, each situation contains 3 items indicating the degree of anxiety expectations (e.g., “*How NERVOUS would you feel about whether or not the teacher will choose you to meet the famous guest*”), angry expectations (e.g., “*How MAD would you feel about whether or not the teacher will choose you to meet the famous guest*”), and feeling disliked (e.g., “*Do you think the teacher meant YOU when she said there was a kid she didn’t like having in the class?*”). The items were rated according to a six-point Likert-type scale (anxious expectations: 1  =  “not nervous,” 6  =  “very, very nervous”; angry expectations: 1  =  “not mad,” 6  =  “very, very mad”; feeling disliked: 1  =  “YES!!!,” 6  =  “NO!!!”). The anxiety expectation, anger expectation, and feeling disliked components were averaged across the twelve instances. Higher scores indicated greater rejection sensitivity. CRSQ has shown strong reliability in adolescents in previous studies (α  =  0.82) [63]. In the present study, Cronbach’s α was 0.83 (angry expectations), α = 0.73 (anxious expectations), and α = 0.80 (total score). The one-factorial structure of the CRSQ was confirmed, *χ*^2^_(252)_ = 728, *p* < 0.001, CFI = 0.95, RMSEA [90%CI] = 0.05 [0.09, 0.11].

*Depressive Symptoms.* Depressive symptomatology was assessed using children’s depression inventory (CDI; [64,65]). The CDI is a reliable and widely used 27-item self-report questionnaire that captured cognitive, affective, and behavioural signs of depressive symptoms in children and adolescents in relation to the previous two weeks. Participants are asked to endorse on a 3-point Likert-scale (0 = absence of symptoms to 2 = definite symptoms) the statement that best describes their behavioural or emotional state with regard to a specific symptom of depression (e.g., “I feel like crying every day”). Items are then summed to a total severity scale in which higher scores represent more severe depressive symptoms (0 to 54). The internal reliability of the scale has been reported high in previous studies with both international [66] and Greek samples [67]. The reliability of the scale for this study was high (α = 0.80). The single-factor structure of the instrument was confirmed, χ^2^(299) = 425, *p* < 0.001, CFI = 0.96, TLI = 0.95, RMSEA [90%CI] = 0.05 [0.04, 0.06].

*Social acceptance* was assessed using self-perception profile for adolescents (SPPA) [68]. The scale has 45 items which tap general self-worth and eight specific domains including: (1) athletic competence, (2) physical appearance, (3) social acceptance, (4) close friendship, (5) romantic appeal, (6) behavioural conduct, (7) job competence, and (8) scholastic competence. For the purposes of the current study, we used only the subscale “Social Acceptance” which consists of 5 statements such as “*Some kids find it difficult to make friends*” and “*Some kids are popular among their peers*.” The SPPA employs a structured alternate format designed to avoid socially desirable responses. The participant first decides which statement is most true for her or him, and then proceeds to decide whether the statement is ‘really true’ or ‘sort of true’. The score for each statement ranges from 1 (lowest) to 4 (highest). The items are presented as pairs of statements contrasting two types of adolescents on one characteristic. Higher scores denoted greater perceived social acceptance by peers. The reliability of the scale for this study was adequate (α = 0.70).

## 3. Results

### 3.1. Gender Differences

In order to examine whether there are statistically significant differences between males and females with respect to the variables under study, the test of means was performed using the *t*-test for independent samples. As shown in Table 1, in the total RS (CRSQ_Total), while scores were slightly higher for girls (*M* = 15.61, *SD* = 7.05) when compared to boys (*M* = 1.33, *SD* = 6.57), this difference did not appear to be statistically significant (*p* > 0.05). Also, the differences in the scale of angry expectations in rejection (CRSQ_Angry) also appear to be statistically non-significant (*p* = 0.592), with boys’ scores (*M* = 6.65, *SD* = 3.17) being slightly higher than girls’ (*M* = 6.45, *SD* = 3.24). In contrast, in terms of anxious expectations of RS (CRSQ_Anxious), there appears to be a statistically significant difference in means between the genders, with girls (*M* = 9.21, *SD* = 4.40) showing more anxious expectations [*t* (286) = −2.86, *p* < 0.01], compared to boys (*M* = 7.79, *SD* = 3.94).

According to Table 1, the girls in the sample seem to have a higher score on the depressive symptoms scale (CDI) (*M* = 21.15, *SD* = 6.79) compared to the boys (*M* = 18.21, *SD* = 5.74), with the above difference being statistically significant [*t* (273) = 2.43, *p* < 0.05]. Finally, the SPPA subscale “Perceptions of peer relationships” also showed a statistically significant difference [*t* (283) = −3.89, *p* < 0.01], with boys having a higher score (*M* = 2.89, *SD* = 0.54), compared to girls (*M* = 2.73, *SD* = 0.65).

### 3.2. Correlations

Pearson *r* correlationsamong all variables are presented in Table 2. More specifically, depression appears to be positively related to overall RS, to anxious rejection expectations, and to angry expectations of rejection. It also appears that perceived social acceptance in peer relationships is negatively related to all other survey variables, depressive symptoms, total RS, angry and, slightly more so, anxious expectations in RS.

### 3.3. Indirect Effect Analysis

Adolescents’ self-perceptions of their peer relationships were then examined as a mediating variable between: (A) overall RS and depressive symptoms, (B) anxious expectations and depressive symptoms, and (C) angry expectations of rejection and depressive symptoms, using the macro PROCESS in SPSS. In more detail:Indirect effect analysis (see Figure 1) showed that overall RS has a significant overall effect on depressive symptoms [c: *B* = 0.323, *SE* = 0.054, *CI* (0.216, 0.429)]. With the effect of perceived social acceptance on peer relationships as a mediator [a: *B* = −0.032, *SE* = 0.005, *CI* (−0.042, −0.022)]; [b: *B* = −2.917, *SE* = 0.635, *CI* (−4.168, −1.666)] the direct effect of total RS on depressive symptoms decreased [c’: *B* = 0.229, *SE* = 0.056, *CI* (0.119, 0.340)]. The indirect effect of total RS on depressive symptoms through perception on peer relationships [*B* = 0.093, *SE* = 0.027, *CI* (0.048, 0.153)] denotes partial indirect effect by the mediator (peer relationships).Indirect effect analysis (see Figure 2) shows that although anxious expectations of RS have a significant effect on the occurrence of depressive symptoms [c: *B* = 0.503, *SE* = 0.087, *CI* (0.333, 0.676)], after controlling for the effect of adolescents’ perceptions of peer relationships as a mediator [a: *B* = −0.059, *SE* = 0.008, *CI* (−0.074, −0.043)]; [b: *B* = −2.877, *SE* = 0.656, *CI* (−4.168, −1.587)], the direct effect of anxious RS on depressive symptoms is reduced [c’: *B* = 0.335, *SE* = 0.093, *CI* (0.152, 0.518)]. The indirect effect of anxious rejection expectations on depressive symptoms shows a large indirect effect of perceived social acceptance on peer relationships [*B* = 0.169, *SE* = 0.045, *CI* (0.090, 0.269)].Finally, the indirect effect analysis with angry rejection expectations as an independent variable (see Figure 3) showed that the overall effect of angry RS was statistically significant [c: *B* = 0.613, *SE* = 0.118, *CI* (0.380, 0.845)]; however, after controlling for the effect of perceived social acceptance on peer relationships as a mediator [a: *B* = −0.045, *SE* = 0.011, *CI* (−0.067, −0.043); [b: *B* = −3.345, *SE* = 0.602, *CI* (−4.530, −2.160)], the direct effect of the independent variable (angry expectations of rejection) on depressive symptoms decreases [c’: *B* = 0.463, *SE* = 0.115, *CI* (0.237, 0.690)]. The indirect effect of angry rejection expectations through children’s peer relationships on depressive symptoms [*B* = 0.149, *SE* = 0.052, *CI* (0.065, 0.276)] suggests a statistically significant and strong indirect effect.

As gender differences were observed in anxious rejection sensitivity and depressive symptoms, moderation analysis was conducted in this relationship. Results showed that gender did not moderate the effect of anxious rejection in depressive symptomatology of our sample [*B* = −0.086, *t* = −0.574, *BCa CI* (−0.381, 0.209)].

## 4. Discussion

The purpose of the present study was to investigate the indirect effect of perceived peer acceptance in the relationship between RS and depressive symptoms in a sample of adolescent males and females.

Starting with gender differences in scale scores, girls showed statistically higher scores than boys on anxious rejection expectations but not on angry rejection expectations. In the total RS scale, girls’ scores were increased, but not statistically significant, while in angry RS, boys had higher scores, which were also not significant. The present findings are consistent with the existing literature where there seems to be confusion and disagreement as to the existence of gender differences in rejection susceptibility, and all sorts of hypotheses have been put forward. In research by Harper and colleagues [69], there have not appeared to be any gender differences in RS, while other research has suggested that boys are likely to be more sensitive to rejection in late adolescence [49], but this does not appear to be confirmed in the present research. However, a larger body of research shows that sensitivity to rejection is higher in the female population [8,70,71]. On the other hand, girls’ higher scores in anxious RS can be justified to some extent, as anxious RS is directly related to internalizing problems [72], which are more common in women, whereas aggression and general externalizing problems are observed in adolescent boys as precursors to depression in adulthood [73], which may partly explain the slightly higher scores of boys in angry RS. Consistent with the above findings—of increased internalizing problems in girls—are the findings for scores on the CDI scale, where girls scored statistically higher in depressive symptomatology than boys. As mentioned above, it has been suggested that girls experience twice the rate of depressive symptoms as early as adolescence, a difference that continues into later adulthood [74,75]. Finally, a significant gender difference was also observed in the level of perceived social acceptance by peers, with boys scoring higher, suggesting that boys appear to perceive their peer relationships as more positive and accepting than girls. This finding can probably be explained by the fact that girls rate themselves more negatively in terms of social competence [37] and report more negative interpersonal stress in the peer domain [76]. This combined with girls’ increased orientation towards social peer relationships from as early as early adolescence [77] may make them more vulnerable to interpersonal difficulties [78] and therefore more susceptible to identifying and perceiving their relationships as more rejecting.

The central aim of the present study was to examine the indirect effect of perceived social acceptance in peer relationships on the relationship between RS and depressive symptoms. Therefore, in addition to the statistically significant correlations of all the survey variables with each other, the indirect effect analysis also showed statistically significant results for the role of the mediating variable, which confirms the initial expectations of the study.

In contrast, the mediation analysis showed that anxious and angry RS had a fairly high and significant overall effect on depressive symptoms, which decreased after the effect of perceived social acceptance, indicating in both cases an indirect effect. More specifically, we observe that while the largest direct effect on the dependent variable is shown by angry RS, the largest indirect effect is observed in the model with anxious RS as the independent variable. In another longitudinal study [79] on a large sample of adolescents (*N* = 6.504), the mediating role of perceptions of peer group relationships on the relationship between anxiety in adolescence and depression levels in adulthood (12 to 14 years later) was shown.

As noted earlier in this paper, negative cognitive biases have been implicated in combination with RS as risk factors for depression [17], and these biases may be considered a potential mechanism linking anxious RS to depressive symptoms [80]. In short, anxious anticipation of rejection, and RS more generally, may lead to negative interpretations of social situations, such as, for example, the perception of peer relationships as rejecting, resulting in a self-perpetuating cycle of negative responses to the behaviour of others and negative social interactions, which in turn may lead to the emergence of negative emotions, as presented in depression [81].

However, beyond the impact of anxious rejection expectations on the occurrence of depressive symptoms and the mediating model, the equally important impact that angry rejection expectations have on depressive symptoms should not be ignored. The results of our study showed that the direct effect of angry expectations on depressive symptoms is very high with the same being true for the indirect effect of peer acceptance perceptions in this relationship. This high effect can probably be explained by the fact that the constant anticipation of rejection, whether associated with angry or anxious reactions, creates feelings of anxiety that promote depressive symptoms. Also, depressive symptoms and aggressive behaviour have been found to be closely related in adolescents and girls [19,21], while anger and aggression are associated with childhood and adolescent depression [9,22].

### Advantages, Limitations, and Future Research Recommendations

This research, like any research, has both limitations and advantages. Firstly, regarding the characteristics of the sample, despite the fact that it is a sufficient number of sample (*N* = 295) to be able to provide reliable results, the participants came only from a specific region of Crete, namely the Prefecture of Heraklion, which makes the sample not representative of the total population. The lack of racial/ethnic/religious/cultural diversity in the current research potentially limits the generalizability of findings to a broader population, however, pointing to the need for greater diversity and inclusion in psychological research. Another limitation of the survey concerns the process of completing the questionnaires. The large size of the questionnaires and the time needed to complete them (about 40 min) may have been a factor of fatigue that affected the concentration and interest of the participants. Furthermore, the cross-sectional nature of the data does not allow for any causal relationships among study variables. It would be useful to conduct longitudinal studies examining the way in which adolescents function in their relationships longitudinally and the developmental pathways of the perception of rejection from their peers. Possible moderating (i.e., buffering) factors could be included in a similar longitudinal model such as resilience, self-regulation, and friendship quality. For example, it could be tested whether adolescents who regulate their emotions in a more efficient way are protected against developing depressive symptomatology, although they exhibit elevated scores in RS. Finally, this study was not able to take into account possible emotional states such as irritability which has been debated in the relevant literature as a potential contributing factor of RS [82].

Nevertheless, the present study also has important advantages. One advantage is the contribution to counselling practice, as theresults can be appropriately utilised in counselling adolescents by aiming to prevent and reduce depressive symptoms during this developmental stage. The results of this study highlight the role of rejection sensitivity and the importance of adolescents’ perceptions of their peer relationships in the occurrence of depressive symptoms. Counselling interventions would therefore be appropriate to focus on educating and strengthening the social and communication skills of vulnerable children, aiming to develop positive peer friendships. Also, interventions aimed at improving the individual’s ability to manage emotions such as anger and anxiety resulting from sensitivity to rejection, and interventions aimed at enhancing the child’s self-esteem and self-worth, which is affected by rejection in peer relationships [83,84,85], could help in this direction. Therefore, focusing on and reducing the impact of these two variables, both by educational staff, family, and mental health professionals, will help prevent and/or reduce the phenomenon of depression in this sensitive period. The implementation of screening tests in detecting anxiety and/or depressive symptomatology within the school settings and the appropriate referral to counselling services of students at risk might be an option towards this direction.

## 5. Conclusions

In summary, the present study identifies and confirms the indirect role of perceived social acceptance in peer relationships in the relationship between RS and depressive symptoms in adolescents, confirming the importance of close relationships with peers during this period. Finally, differences were observed in the models between the components of RS (angry and anxious RS expectations), with the indirect effect of perceptions being higher in the model with anxious RS, which seems to have theoretical support from previous studies (e.g., [80]).

## Figures and Tables

**Figure 1 behavsci-14-00010-f001:**
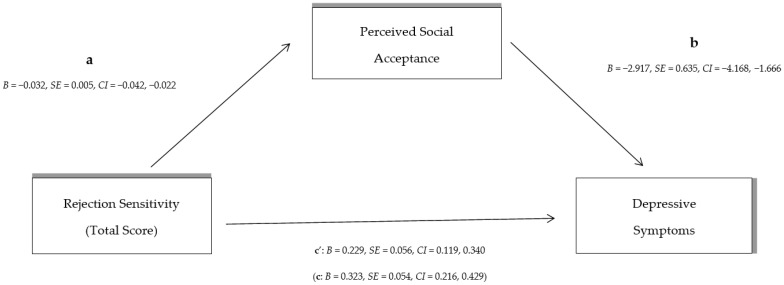
The indirect effect of total rejection sensitivity on depressive symptoms through perceived social acceptance by peers (the direct total effect of rejection sensitivity on depressive symptoms in parenthesis).

**Figure 2 behavsci-14-00010-f002:**
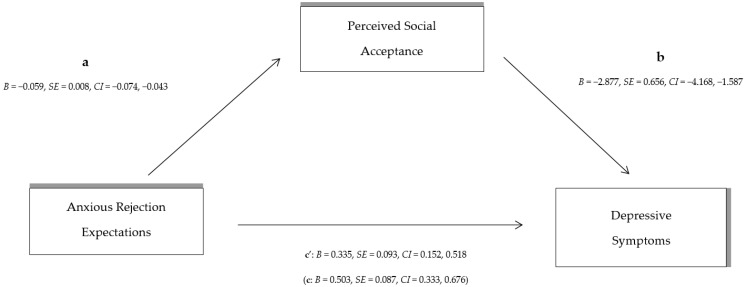
The indirect effect of total anxious rejection sensitivity on depressive symptoms through perceived social acceptance by peers (the direct total effect of anxious rejection sensitivity on depressive symptoms in parenthesis).

**Figure 3 behavsci-14-00010-f003:**
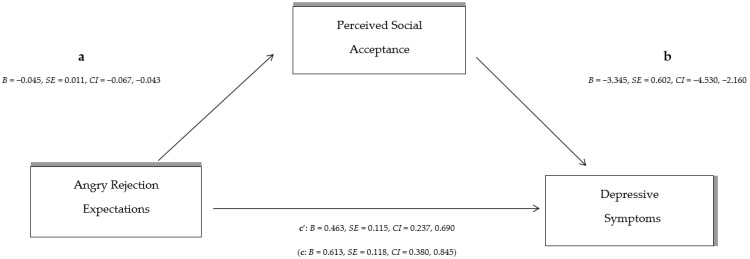
The indirect effect of total angry rejection sensitivity on depressive symptoms through perceived social acceptance by peers (The direct total effect of angry rejection sensitivity on depressive symptoms in parenthesis).

**Table 1 behavsci-14-00010-t001:** Means, standard deviations gender differences, and reliabilities of the study variables.

Gender							
	Males		Females		*t*-Test		Cronbach’s *a*
	Mean	SD	Mean	SD	t	df	
CRSQ_Total	14.33	6.57	15.61	7.05	ns	285	0.80
CRSQ_Anxious	7.79	3.94	9.21	4.40	−2.86 **	286	0.83
CRSQ_Angry	6.65	3.17	6.45	3.24	ns	293	0.73
CDI	18.21	5.74	21.15	6.79	−3.89 **	273	0.80
SPPA_SA	2.89	0.54	2.73	0.65	2.43 *	287	0.70

* *p* < 0.05. ** *p* < 0.01. CRSQ_Total = Rejection Sensitivity total score, CRSQ_Anxious = anxious rejection expectations, CRSQ_Angry = angry rejection expectations, CDI = depressive symptoms, SPPA_SA = Perceived social acceptance by peers.

**Table 2 behavsci-14-00010-t002:** Pearson *r* correlations among study variables.

Pearson (*r*)	1	2	3	4	5
1. CRSQ_Total	-				
2. CRSQ_Anxious	0.943 **	-			
3. CRSQ_Angry	0.896 **	0.698 **	-		
4. CDI	0.353 **	0.343 **	0.312 **	-	
5. SPPA_SA	−0.351 **	−0.403 **	−0.230 **	−0.368 **	-

** *p* < 0.01. CRSQ_Total = Rejection Sensitivity total score, CRSQ_Anxious = anxious rejection expectations, CRSQ_Angry = angry rejection expectations, CD I = depressive symptoms, SPPA_SA = Perceived social acceptance by peers.

## Data Availability

The data presented in this study are available on request from the corresponding author.

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
