# Peer review of "The Relationship of Rejection Sensitivity to Depressive Symptoms in Adolescence: The Indirect Effect of Perceived Social Acceptance by Peers"

_behavsci, 2023, doi:10.3390/bs14010010_

Round 1

Reviewer 1 Report

Comments and Suggestions for Authors

Summary: 

Abstract

Specific comments:

Background (Lines 9-13): Consider adding why focusing on Greek adolescents is important—does it add a unique cultural angle or generalize the problem?

Methods (Lines 15-18): Demographic information is useful. Briefly mention the types or names of the self-report questionnaires for better context.

Results (Lines 19-25): Well-summarized, but the multiple elements being studied (RS, depressive symptoms, peer relationships, gender differences) could make it complex for the reader. Consider splitting into bullets to delineate the different result categories, especially to separate the part about gender differences and types of RS.

Introduction

(Optional) Overlap with Other Disorders: This section could benefit from a discussion on the comorbidity or overlap of rejection sensitivity and depression with other disorders like anxiety or ADHD.

Results:

Potential Multicollinearity: The variables CRSQ_Anxious and CRSQ_Angry have a high correlation. If these are used in the same regression model, they might introduce multicollinearity, which is not discussed. The paper would benefit from mentioning this; perhaps for a future study or as a limitation. 

Discussion

The paper at times switches between terms like "RS", "anxious RS", "angry RS", and "rejection sensitivity" with little setup or differentiation. Readers who aren't specialists in the field may get lost. Make sure that abbreviations are clearly defined upon first use and that you're consistent with terminology throughout.

(Missing sections that help to add clarity)

Limitations: The limitations focus mainly on the sample size and methodology, but it would be useful to include intellectual limitations. For example, is there a current debate in the field that this study is not able to settle? If so, that could be considered a limitation.

Future Research: This part is often neglected but extremely important. While you mention the necessity of longitudinal studies, you might want to provide more specifics. What kinds of research questions should future studies aim to answer? Which methodologies might be most appropriate for answering them?

Author Response

Dear Reviewer,

We appreciate the time and effort that you have dedicated to providing feedback on our manuscript and are grateful for the insightful and valuable comments that have improved our work. We have addressed all of your suggestions, and we hope they would be at a satisfactory level. Those changes are highlighted within the manuscript. Please see below, for a point-by-point response to your comments.

Abstract

Specific comments:

  1. Background (Lines 9-13): Consider adding why focusing on Greek adolescents is important—does it add a unique cultural angle or generalize the problem?

Thank you for this comment. We added this paragraph accordingly (lines 164-170): “It should be noted here that depression symptomatology represents a major health issue for Greek adolescents as its prevalence has been found to be higher compared to similar age groups across many European countries (57). Further, Greece has entered a long period of economic crisis, especially after the break of covid-19 pandemic with major adverse effects on many areas of the life of the population. We therefore consider as important the investigation of some significant mental health aspects of Greek adolescents during this difficult period.  

  1. Methods (Lines 15-18): Demographic information is useful. Briefly mention the types or names of the self-report questionnaires for better context.

            We have included this information

  1. Results (Lines 19-25): Well-summarized, but the multiple elements being studied (RS, depressive symptoms, peer relationships, gender differences) could make it complex for the reader. Consider splitting into bullets to delineate the different result categories, especially to separate the part about gender differences and types of RS.

            We have made these changes accordingly

Introduction

  1. (Optional) Overlap with Other Disorders: This section could benefit from a discussion on the comorbidity or overlap of rejection sensitivity and depression with other disorders like anxiety or ADHD.

We have included this paragraph accordingly (lines 65-70): Downey et al. [13] have stressed that Anxious RS would be more strongly linked with internalizing difficulties such as depressive symptomatology and social anxiety, whereas angry RS would be more strongly associated with externalizing difficulties, such as aggressive behavious. Further, Anxious RS has been linked with other psychological difficulties such as social anxiety and ADHD, especially in adolescent samples [13]

Results:

  1. Potential Multicollinearity: The variables CRSQ_Anxious and CRSQ_Angry have a high correlation. If these are used in the same regression model, they might introduce multicollinearity, which is not discussed. The paper would benefit from mentioning this; perhaps for a future study or as a limitation. 

Thank you for this comment, however we did not include these variables (which indeed have a high correlation as they are components of a common construct) in the same regression model. Rather, we run separate mediation models with each one of them as an independent variable, accordingly.

Discussion

  1. The paper at times switches between terms like "RS", "anxious RS", "angry RS", and "rejection sensitivity" with little setup or differentiation. Readers who aren't specialists in the field may get lost. Make sure that abbreviations are clearly defined upon first use and that you're consistent with terminology throughout.

We have used the same abbreviation throughout the manuscript for better clarity.

  1. Limitations: The limitations focus mainly on the sample size and methodology, but it would be useful to include intellectual limitations. For example, is there a current debate in the field that this study is not able to settle? If so, that could be considered a limitation.

Thank you for this comment; we have added this sentence in the Limitation section:  “Finally, this study was not able to take into account possible emotional states such as irritability which has been debated in the relevant literature as potential contributing factor of RS” (Quarmley et al., 2023)

  1. Future Research: This part is often neglected but extremely important. While you mention the necessity of longitudinal studies, you might want to provide more specifics. What kinds of research questions should future studies aim to answer? Which methodologies might be most appropriate for answering them?

We feel that we do provide some relevant ideas when we mention “Possible moderating (i.e. buffering) factors could be included in a similar longitudinal model such as resilience, self-regulation and friendship quality”. However, we added this sentence: “For example, it could be tested whether adolescents who regulate their emotions in a more efficient way are protected against developing depressive symptomatology although they exhibit elevated scores in RS.

Reviewer 2 Report

Comments and Suggestions for Authors

Thank you for the opportunity to revise this manuscript.

This manuscript is convincingly ambitious, sound, credible, and has soundness and credible methodology.

- The topic of this manuscript is interesting, and it introduces an innovative aspect.  The theme is relevant and has a conceptual domain, the study is a well-conceived, well-crafted, and well-presented paper.

- The methods are appropriate, accurate, and objective for the experiments, and improve the understanding of the reader.

- The analyses are while not complex, informative, and appropriate for the experiments.

- The authors recognize the limitations of this research, and in my view, this manuscript has main value and provides evidence that the authors seem to recognize the main innovation of their study,

- Overall, the paper is well-placed to stimulate future research within the field.

 - It is clear a substantial amount of work has gone into preparing this manuscript, and it can be reconsidered able to be published and may result in a modest contribution to the literature, so I recommend its rapid publication after minor revisions.

Reformulation of the abstract: for example this sentence: "Conclusions: The implications of the findings for the development of effective therapeutic interventions are also discussed" does not make sense. authors should include clear conclusions

Methods: authors must reformulate the subsection of instruments. When, for example, they put the following sentence: "The reliability of the scale for this study was good (α=.70)", they must justify why it is considered good. Either they are descriptive and present only the total alpha values for the study or if they want to infer they have to justify the interpretation.

In the discussion: the authors present a sub-topic entitled: "4.1. Advantages, limitations and future research recommendations" that should reformulate and make it stronger.

The authors should also include the Conclusions Topic and leave a strong impression on the main conclusions of the study.

I  hope my comments will be useful in the process of revising this manuscript.

Author Response

Dear Reviewer,

We appreciate the time and effort that you have dedicated to providing feedback on our manuscript and are grateful for the insightful and valuable comments that have improved our work. We have addressed all of your suggestions, and we hope they would be at a satisfactory level. Those changes are highlighted within the manuscript. Please see below, for a point-by-point response to your comments.

  1. Reformulation of the abstract: for example this sentence: "Conclusions: The implications of the findings for the development of effective therapeutic interventions are also discussed" does not make sense. authors should include clear conclusions

We reformulated this part of the Abstract as follows: “The results of this study highlight the role of rejection sensitivity and the importance of adolescents' perceptions of their peer relationships in the occurrence of depressive symptoms during this developmental period.”

  1. Methods: authors must reformulate the subsection of instruments. When, for example, they put the following sentence: "The reliability of the scale for this study was good (α=.70)", they must justify why it is considered good. Either they are descriptive and present only the total alpha values for the study or if they want to infer they have to justify the interpretation.

- Thank you for this comment. We changed “good” to “adequate” in order to be more clear.

  1. In the discussion: the authors present a sub-topic entitled: "4.1. Advantages, limitations and future research recommendations" that should reformulate and make it stronger.

- Thank you for this comment. We have added this paragraph in this sub-topic: ‘For example, it could be tested whether adolescents who regulate their emotions in a more efficient way are protected against developing depressive symptomatology although they exhibit elevated scores in RS. Finally, this study was not able to take into account possible emotional states such as irritability which has been debated in the relevant literature as potential contributing factor of RS”

  1. The authors should also include the Conclusions Topic and leave a strong impression on the main conclusions of the study.

- Following your suggestion, we incorporated a Conclusion sub-section at the end of the manuscript.

Reviewer 3 Report

Comments and Suggestions for Authors

This study aimed to examine the relationship between Rejection sensitivity (RS)and depressive symptoms with a sample of Greek adolescents, with a particular interest in the mediating role of perceived social acceptance by peers as well as the possible gender differences. This study is generally within the scope of this journal and with appropriate writing and structuring. However, some points need to be addressed to further proceed the manuscript. First, more elaboration on the indirect effect of perceived social acceptance is needed, as the relationship between RS and perceived social acceptance tend to be reciprocal rather than unidirectional. Second, research hypotheses should be listed respectively following each literature review section to guide the writing of the manuscript in response to the research questions. Third, the construct validity of the measures should be reported, and the reasons for not using more advanced structural equation modeling to deal with the path analysis should be clarified. Finally, the practical implications corresponding to the research findings should be given in detail. 

Comments on the Quality of English Language

Minor improvement is required

Author Response

Dear Reviewer,

We appreciate the time and effort that you have dedicated to providing feedback on our manuscript and are grateful for the insightful and valuable comments that have improved our work. We have addressed all of your suggestions, and we hope they would be at a satisfactory level. Those changes are highlighted within the manuscript. Please see below, for a point-by-point response to your comments.

  1. First, more elaboration on the indirect effect of perceived social acceptance is needed, as the relationship between RS and perceived social acceptance tend to be reciprocal rather than unidirectional.

Thank you for this comment. We feel that we justify the indirect effect of perceived social acceptance in the paragraph “As previously reported…is also examined”. However, we added a sentence in this paragraph for better clarity: “For this theoretical reason we included RS as an independent variable in our model that would predict perceived social acceptance".

  1. Second, research hypotheses should be listed respectively following each literature review section to guide the writing of the manuscript in response to the research questions.

- We mention our research hypotheses after discussing the relevant literature review (e.g. line 88 “Therefore, it is assumed in the present research that angry RS could equally predict depressive symptoms”, line 160 “Because of these mixed findings, the role of gender was explored in all analyses without any specific hypotheses”, line 169 “For this reason, in addition to the impact that susceptibility to rejection appears to have on the onset of depressive symptoms, the indirect effect of adolescents' perceived peer acceptance on this relationship is also examined”

  1. Third, the construct validity of the measures should be reported, and the reasons for not using more advanced structural equation modeling to deal with the path analysis should be clarified.

- Thank you for this comment. As for the construct validity of the measures, this was not deemed necessary to be included in this manuscript for two reasons: a) it was not within the scope of the study and b) we used measures that have been already adapted and tested for their construct validity in Greek adolescent samples (see measures’ references)

- We used the computational tool PROCESS as it simplifies the implementation of mediation analysis with observed variables as in our study. We based our decision on Hayes et al’s proposition that: “However, for models of observed variables (i.e., nothing latent), differences in results tend to be trivial, and rarely will the substantive conclusions a researcher arrives at be influenced by the decision to use PROCESS rather than SEM… For models that are based entirely on observed variables, investigators can rest assured that it generally makes no difference which is used, as the results will be substantively identical”

 see Andrew F. Hayes, Amanda K. Montoya, Nicholas J. Rockwood (2017). The analysis of mechanisms and their contingencies: PROCESS versus structural equation modeling Australasian Marketing Journal (2017), doi: 10.1016/j.ausmj.2017.02.001

  1. Finally, the practical implications corresponding to the research findings should be given in detail. 

Although we feel that we have included this information in the last paragraph of sub-section 4.1. “The results of this study…sensitive period” we added this sentence: “The implementation of screening tests in detecting anxiety and/or depressive symptomatology within the school settings and the appropriate referral of students at risk might be an option towards this direction.

Reviewer 4 Report

Comments and Suggestions for Authors

The purpose of the present study was to investigate the indirect effect of perceived peer acceptance in the relationship between rejection sensitivity and depressive symptoms in a sample of adolescent males and females.

Overall the paper is clearly and well written.

I report below some suggestions.

Line 12: the authors should add “: a)” before “the indirect effect of”

Sample: why the authors emphasize the parents’ degree of education?

Procedure: the authors should clarify how they collected parental consent, pointing out the specific procedure. They should also specify inclusion and exclusion criteria.

Measures: the authors should specify how they categorized “family status”

Resutls and Discussion: as the authors used some demographic variables, why did they not analyze all the variables? They seem to have focused only on gender differences. I think they could analyze also the other variable. Or they should motivate this choice.

Author Response

Dear Reviewer,

We appreciate the time and effort that you have dedicated to providing feedback on our manuscript and are grateful for the insightful and valuable comments that have improved our work. We have addressed all of your suggestions, and we hope they would be at a satisfactory level. Those changes are highlighted in green within the manuscript. Please see below, for a point-by-point response to your comments.

  1. Line 12: the authors should add “: a)” before “the indirect effect of”

We added this accordingly

  1. Sample: why the authors emphasize the parents’ degree of education?

We removed this information from the Sample subsection as indeed we did not use this information in any further analyses.

  1. Procedure: the authors should clarify how they collected parental consent, pointing out the specific procedure. They should also specify inclusion and exclusion criteria.

Thank you for your comment; We included this information in page 7 “A sealed letter…in the study”.

  1. Measures: the authors should specify how they categorized “family status”

We removed “family status” from the measures subsection as it was a typo mistake.

  1. Resutls and Discussion: as the authors used some demographic variables, why did they not analyze all the variables? They seem to have focused only on gender differences. I think they could analyze also the other variable. Or they should motivate this choice.

Thank you for your comment. We removed from the description of our sample demographic variables that were not analyzed (e.g. parents’ level of education).

Reviewer 5 Report

Comments and Suggestions for Authors

Thanks for the Editor for giving me this opportunity to review this article. This is an interesting and potentially important study and manuscript. The majority of my comments are directed at ways the manuscript can be improved. The following points need to be further improved:

(1)Abstract: The author should clearly set forth the main viewpoints of this study in the conclusion section.

(2)Introduction: The content is well described and contextualized with respect to previous and present theoretical background and empirical research on the topic, the structure is not very clear. The authors would better add some secondary titles.

(3) Materials and Methods: How is the sample size calculated?What are the inclusion and exclusion criteria for participants?

(4) Results: Since there are gender differences in the main variables, further analysis should be conducted on the moderating effect of gender.

(5) Diversity concerns are not criteria for publication but must be addressed. The nature of the discussion and amount of space devoted to the discussion is the responsibility of the author(s). Diversity includes all aspects of human differences such as but not limited to socioeconomic status, race, ethnicity, language, nationality, sex, gender identity, sexual orientation, religion, geography, ability, age, and culture.

Author Response

Dear Reviewer,

We appreciate the time and effort that you have dedicated to providing feedback on our manuscript and are grateful for the insightful and valuable comments that have improved our work. We have addressed all of your suggestions, and we hope they would be at a satisfactory level. Those changes are highlighted in green within the manuscript. Please see below, for a point-by-point response to your comments.

(1)Abstract: The author should clearly set forth the main viewpoints of this study in the conclusion section.

We reworded this subsection accordingly.

 (2)Introduction: The content is well described and contextualized with respect to previous and present theoretical background and empirical research on the topic, the structure is not very clear. The authors would better add some secondary titles.

Following your suggestion, we added two secondary titles in the Introduction section.

(3) Materials and Methods: How is the sample size calculated? What are the inclusion and exclusion criteria for participants?

Thank you for your comment; following your suggestion, we included this information in the text (page 5).

  (4) Results: Since there are gender differences in the main variables, further analysis should be conducted on the moderating effect of gender.

Following your suggestion, we included the moderation analysis of gender in the relationship between anxious rejection sensitivity and depressive symptoms (page 9).

(5) Diversity concerns are not criteria for publication but must be addressed. The nature of the discussion and amount of space devoted to the discussion is the responsibility of the author(s). Diversity includes all aspects of human differences such as but not limited to socioeconomic status, race, ethnicity, language, nationality, sex, gender identity, sexual orientation, religion, geography, ability, age, and culture.

Thank you for your comment; following your suggestion, we included this information in the text (page 10).

Reviewer 6 Report

Comments and Suggestions for Authors

Dear Author;

In terms of strengths, we have:

- There is a good literature review directly related to the research topic.

- References included in the document are properly cited.

- The research objective is adequately reflected.

- The sample from which the information has been obtained is significant.

- The methodology is clear, adequate, and relevant, making it easy to support the sample information adequately.

- The analysis and results of this research are adequately captured and give good support and value to the research.

- A discussion is made in which the information from this research is contrasted with the work of other authors.

In relation to its areas for improvement, I can mention:

- It is suggested to adjust the Abstract to the journal's requirement, where they ask for an IMRAD structure without using headlines. On the other hand, paragraph a) of the study objectives is not included.

- Finally, the conclusion is terse, mentioning that "the results obtained have theoretical support from previous studies", in which case it would be useful to mention one of these studies.

- In short, it is a good work that needs a few suggestions for revision.

Best regards 

Author Response

Dear Reviewer,

We appreciate the time and effort that you have dedicated to providing feedback on our manuscript and are grateful for the insightful and valuable comments that have improved our work. We have addressed all of your suggestions, and we hope they would be at a satisfactory level. Those changes are highlighted in green within the manuscript. Please see below, for a point-by-point response to your comments.

  1. It is suggested to adjust the Abstract to the journal's requirement, where they ask for an IMRAD structure without using headlines. On the other hand, paragraph a) of the study objectives is not included.

Thank you for your comment; following your suggestion, we removed headlines from the Abstract.

  1. Finally, the conclusion is terse, mentioning that "the results obtained have theoretical support from previous studies", in which case it would be useful to mention one of these studies.

 Following your suggestion, we referenced this sentence.

Round 2

Reviewer 3 Report

Comments and Suggestions for Authors

The questions as mentioned in the first round were not well addressed. 

Author Response

  1. Following the suggestion of the reviewer’s first comment, we elaborated in more detail the indirect effect of perceived social acceptance (page 4, paragraph 3).
  2. Following the suggestion of the reviewer’s third comment, we included the Confirmatory Factor Analysis of the study measures (page 6). We did not include the CFA of the Social Acceptance measure as it consists a subscale of a broader instrument (SPPA) as mentioned in the manuscript.

Reviewer 5 Report

Comments and Suggestions for Authors

I am satisfied with the author's response to my comments. However, there is still a minor issue. Since the theoretical sample size is 140, why is the actual sample size 295? The author can provide a brief explanation in the manuscript.

Author Response

  1. "I am satisfied with the author's response to my comments. However, there is still a minor issue. Since the theoretical sample size is 140, why is the actual sample size 295? The author can provide a brief explanation in the manuscript."

Response: Thank you for your comment, we have provided a brief explanation for this choice (highlighted in blue) within the manuscript.

Round 3

Reviewer 3 Report

Comments and Suggestions for Authors

The authors have effectively addressed my concerns regarding the manuscript.

Author Response

  1. "The authors have effectively addressed my concerns regarding the manuscript."Dear Reviewer, Thank you for your valuable comments which we believe have improved the scholarship of this manuscript